# Diagnosis of Mitral Valve Prolapse: Much More than Simple Prolapse. Multimodality Approach to Risk Stratification and Therapeutic Management

**DOI:** 10.3390/jcm11020455

**Published:** 2022-01-17

**Authors:** Ali Alenazy, Abdalla Eltayeb, Muteb K. Alotaibi, Muhammah Kashif Anwar, Norah Mulafikh, Mohammed Aladmawi, Olga Vriz

**Affiliations:** 1Heart Centre, King Faisal Specialist Hospital & Research Centre, Riyadh 11211, Saudi Arabia; dr_alialenazy@yahoo.com (A.A.); abdullaheltayeb2002@gmail.com (A.E.); amuhammadkashif@kfshrc.edu.sa (M.K.A.); madmawi@kfshrc.edu.sa (M.A.); 2Radiology Department, King Faisal Specialist Hospital & Research Centre, Riyadh 11211, Saudi Arabia; MM99MM9911@HOTMAIL.COM (M.K.A.); Mulafikhnoura@gmail.com (N.M.)

**Keywords:** mitral valve prolapse, mitral annulus disjunction, myocardial fibrosis, complex ventricular arrhythmias, cardiac magnetic resonance, echocardiography

## Abstract

Mitral valve prolapse (MVP) is the most common valvular disease with a prevalence of 2%. It has generally a benign course; however, recent findings suggested an association between MVP and complex arrhythmias and eventually cardiac arrest and for this reason, it is also called arrhythmogenic MVP. Subjects who experience this complication are in general young women, with thickened mitral leaflets or bileaflet prolapse not necessarily associated with severe mitral regurgitation (MR). The nature of the relation between MVP and cardiac arrest is not clearly understood. Actually, the challenging task is to find the cluster of prognostic factors including T-wave inversion, polymorphic premature ventricular contractions, bileaflet prolapse, MR severity, but most importantly, those parameters of hypercontractility, mitral annulus disjunction (MAD), and myocardial fibrosis using a multimodality approach. Transthoracic echocardiography is the first-line imaging modality for the diagnosis of MVP, but also for detecting MAD and hypercontractility, followed by cardiac magnetic resonance for tissue characterization and detection of myocardial and papillary muscle fibrosis, using either late gadolinium enhancement (at the basal segment of the inferolateral wall and papillary muscles) (macro-fibrosis), or diffuse fibrosis by T1 mapping (native and post contrast T1). Moreover, there are also preliminary data on positron emission tomography utilizing ^18^F-fluorodeoxyglucose as a tool for providing evidence of early myocardial inflammation. The objective of this review article is to provide the clinician with an overview and a practical clinical approach to MVP for risk stratification and treatment guidance.

## 1. Introduction

Mitral valve prolapse (MVP) is the displacement of one or both mitral leaflets into the left atrium in systole, described for the first time by Barlow in 1963 [1]. The diagnosis is made by clinical examination and echocardiography. MVP is the most common valve disease occurring in 1.2–3% of the general population [2,3,4,5]. It is not found in newborns [6] and its prevalence is low among children and adolescents (0.3% and 0.6%, respectively) [7,8], suggesting that MVP is a progressive degenerative disease. Overall MVP is a benign disease, associated with non-specific symptoms, such as atypical chest pain, exertional dyspnea, palpitations, anxiety, mid-systolic click, low blood pressure, and leaner build, which have come to be known as MVP syndrome. MVP can also be associated with ECG abnormalities and complex ventricular arrhythmias (c-VA) with polymorphic/right bundle branch block (RBBB) morphology. The complications associated with valve disease include mitral regurgitation (MR) and less common infective endocarditis and cerebrovascular ischemic events. A subset of patients can experience cardiac arrest or sudden cardiac death (SCD) due to c-VA, which is a dramatic event that can affect these otherwise young healthy subjects, as described by Nishimura in 1985 [9]. The incidence of SCD is estimated to be 0.14–1.8% per year in patients with MVP [4,10] with a prevalence of 2.3% [10], but it was reported to be as high as 7% in a young SCD population (13% of women) [11]. The estimated rate of SCD in the general population is 0.06–0.08%/year, but the risk of SCD in MVP and cardiac patients is 1.73–2.3 times higher.

The renewed interest in MVP derives from anatomo-pathological findings [11,12,13,14] and, more recently, multimodality imaging studies [11,14,15,16,17,18] that identified myocardial lesions responsible for c-VA and eventually SCD. In fact, the so-called malignant “arrhythmic MVP “(AMVP) seems to have a specific phenotype including thickened bileaflets, fibrosis of papillary muscles and inferobasal wall, and mitral annulus disjunction (MAD), a structural abnormality manifesting as a wide separation between the left atrial wall–mitral valve junction and the left ventricular (LV) attachment in the setting of no significant MR [14]. As elegantly described by Reinier and Chugh in their editorial [19] commenting Dilling’s et al. paper [13], postmortem diagnosis alone might underestimate the prevalence of malignant MVP whereas cardiac magnetic resonance (CMR) can have the potential to identify those MVP patients at high risk. This is particularly important because, beyond the classical definition of MVP, the challenge is to identify the subset of patients at risk of developing c-VA and SCD using multimodality imaging and to guide the best treatment (Figure 1).

## 2. Definition of Mitral Valve Prolapse

MVP is defined as the displacement of one or both mitral valve leaflets or part of a leaflet (scallop) of more than 2 mm beyond the mitral annulus into the left atrium in the parasternal long-axis view with or without MR. Classic prolapse is defined as a mitral leaflet thickness of at least 5 mm and non-classic prolapse as a mitral leaflet thickness of less than 5 mm [20]. MVP can be classified as primary (non-syndromic) or secondary (syndromic), the latter being associated with connective tissue disorders such as Marfan syndrome, Loeys-Dietz syndrome, Ehler-Danlos syndrome, osteogenesis imperfecta [21]. Additionally, familial clustering of MVP with an X-linked inheritance pattern and transmission has also been described [21,22]. MVP can also be caused by myxomatous degeneration (Barlow’s disease [BD]) or fibroelastic deficiency (FED), which is characterized by leaflets thinning and focal chordae tendineae elongation or rupture. Although the two diseases share common histological alterations, whether BD and FED are a continuum of the same disease has yet to be defined [23], but the clinical presentation is different. FED patients are older (>60 years), have relatively acute symptoms mostly related to chordae rupture, whereas BD patients are younger, frequently have long history of a cardiac murmur with mid-late systolic click, and usually are asymptomatic. Moreover, patients with FED can suffer from SCD but, differently from BD patients, SCD is likely due to arrhythmias related to severe MR, LV dilatation, and dysfunction [24]. Moreover, as reported by Mantegazza et al. [18] and Essayagh et al. [25], the prevalence of MAD is higher in patients with BD compared to those with FED. Unfortunately, at present, there are no studies comparing CMR features of fibrosis in FED versus BD patients (Figure 2).

MAD is an anatomical entity that has been associated with MVP; it is an abnormal insertion of the hinge line of the posterior mitral leaflet (PML) on the atrial wall, characterized by distinct separation of the mitral valve annulus-left atrial wall and the basal region of the posterolateral LV myocardium, also defined as atrialization of the posterior leaflet base. MAD can involve either a large or small portion of the mitral annulus, usually the mid and central portion. In early reports, this entity was also described in normal hearts, though as a rare finding [12] or as a normal variant [26]. More recently, the group from the University of Padua, Italy showed that MAD and systolic curling of the PML were associated with LV fibrosis of the inferobasal wall and papillary muscles in both SCD victims with MVP and MVP patients with c-VA, likely due to the excessive mobility of the mitral valve apparatus and systolic stretch of the myocardium closely linked to the valve [11,14]. Actually, a pathological cut-off for MAD is not available for any of the different methods used for its detection. In 67 patients undergoing surgical correction of MAD, Eriksson et al. [27] found a mean value of annular disjunction of 10 mm at transesophageal echocardiography (TEE) in severe MVP versus 8.2 mm in a control group with mild to moderate mitral valve degeneration. Dejgaard et al. [15] recorded a mean value of MAD of ≥1 mm on CMR and 6 mm on TTE, and Perazzolo et al. [14] reported a mean value of MAD of 4.8 mm on CMR. Carmo et al. [28] suggested a TTE cut-off of MAD of 8.5 mm with a good sensitivity and specificity for the prediction of non-sustained ventricular tachycardia (VT). It seems therefore that MAD is not an exclusive feature of MVP, but the length of MAD correlates with the severity of prolapse, myocardial fibrosis, and VA.

## 3. Who Are the Patients at Risk of Arrhythmic Complications?

Basso et al. [11] reported for the first time the association of MVP with SCD. Among 650 young adults with SCD (<40 years of age) recorded in the Veneto Region Registry from 1982 to 2013, 43 patients (26 female) with MVP caused by myxomatous valve disease were identified, representing 7% of all SCD. Sudden death occurred during sleep in 81% of cases. Patients had either isolated (PML in 30%) or bileaflet (70%) MVP. When ECG was available (28% of cases) a negative/isodiphasic T wave on the inferior lead was noted and all (100%) had RBBB morphology VA. On inspection, endocardial fibrous plaques on the posterolateral wall were found in 58% of cases. At microscopic examination, LV myocardium showed patchy replacement-type fibrosis at the level of papillary muscles and adjacent free wall in all subjects, and fibrosis at the inferobasal wall under the PML in 88% of patients. In the 30 living patients with MVP and c-VA, VT had LV origin in all and occurred at rest in 87% of cases, and exercise stress test was negative for effort-induced VA. Late gadolinium enhancement (LGE) was localized on the papillary muscles in 83% of patients and on the LV inferobasal segment under the posterior leaflet in 73%. This regional distribution overlapped with autopsy findings in SCD victims. These findings were confirmed by the same authors in MVP patients without moderate to severe MR [14]. The group of patients with MVP and positive LGE included a higher proportion of females, had more c-VA, more frequently bileaflet involvement, longer MAD, and more curling. Interestingly, Guglielmo et al. [29] showed a typical distribution of interstitial fibrosis (T1 mapping) in patients with MVP, but did not find any correlation between fibrosis and MR severity, concluding that this is likely not a direct consequence of volume overload. In a systematic review and meta-analysis published in 2019, Nalliah et al. [4] reported a prevalence of MVP among SCD victims of 1.9%, bileaflet prolapse in 80%, and myocardial fibrosis in 71%. Ventricular ectopy, all with RBBB morphology, was reported in 79%, of which 82% had c-VA. ST-T abnormalities were observed in 65.3% of cardiac arrest victims, including inverted or biphasic T waves in the inferior leads. Moderate to severe MR was reported only in 36.3% of SCD victims with MVP.

On the basis of the aforementioned data, the features of the AMVP phenotype at risk for SCD that need extensive investigation are: (i) young female; (ii) symptoms such as shortness of breath, chest pain, fatigue, anxiety. Particular attention should be paid to lightheaded/dizziness, palpitations, syncope that may be the epiphenomenon of premature ventricular contractions (PVCs), VA, sustained and non-sustained VT; (iii) on auscultation, a mid-systolic click is heard, which represents the abrupt fall and traction from the subvalvular apparatus of the mitral leaflets into the left atrium, with or without late systolic murmur that corresponds to MR; (iv) ECG abnormalities with inverted/biphasic T waves in the inferior leads, Holter ECG findings such as ventricular ectopy with RBBB morphology and c-VA; (v) thickened mitral leaflets and bileaflet prolapse, MAD and curling. In two-thirds of cases, MVP is not associated with significant MR; and (vi) presence of myocardial fibrosis on CMR (Table 1).

## 4. Multimodality Approach in the Identification of Red Flags and Risk Stratification

A multimodality approach is of paramount importance for the risk stratification of MVP patients (Figure 3).

### 4.1. Echocardiography

The first-line imaging modality for the diagnosis of MVP is transthoracic echocardiography (TTE), which allows assessment of the structural characteristics of the valve, i.e., leaflet thickness, uni- or bileaflet prolapse, prolapse size, presence and severity of MR, left atrial and LV dimension and function [30]. Additional focused measurements/parameters need to be evaluated in order to improve the identification of patients at high risk for arrhythmias or SCD [15]. MAD can be seen on TTE in the parasternal long-axis view as well as in the 4- and 2-chamber view [15], along with curling with a good correlation between TTE and CMR, particularly for MAD >2 mm [14,15,31] (Figure 4).

Moreover, curling is suggested to be related to progressive myxomatous degeneration of the mitral leaflets [14]. Prolapse, MAD, and curling are linked to hypercontractility of the basal segments of the inferior, posterior, and anterolateral wall. Myocardial stretch by the prolapsing leaflets results in higher tissue velocity that abruptly affects the papillary muscles and adjacent myocardium; this sharp traction may ultimately be, per se, a trigger of PVCs and c-VA. In addition, long-term hypermobility and stretch could evolve into local fibrosis. Higher tissue velocity can be measured by tissue Doppler imaging from the standard apical window, with the probe aligned according to mitral annulus motion. A Doppler systolic velocity ≥16 cm/s (Pickelhaube sign) is considered a marker of AMVP [32,33]. Hypercontractility is also responsible for the hypertrophy of the basal segments behind the PML [34]. Data derived from CMR in MVP patients showed a basal to mid segment ratio >1.5, which was found to be higher in patients with positive LGE, although without reaching statistical significance [14]. Advanced TTE methods can be used for regional hypercontractility such as segmental longitudinal strain [35] of the mid-basal segments of the posterior and anterolateral walls, and for the detection of increased mechanical work [35,36]. Myocardial work is a parameter that combines longitudinal deformation and afterload, and MVP patients with regional increase in longitudinal strain also have increased mechanical work. The underlying rationale is that repeated local myocardial traction increases energy demand and oxidative stress with local hypertrophy and fibrosis [11] (Figure 5).

Furthermore, Ermakov and colleagues [37] showed that patients with AMVP had more bileaflet prolapse compared to non-AMVP patients and greater mechanical dispersion, a parameter related to arrhythmic events in several other pathologies [38,39].

Transesophageal echocardiography (TEE) provides the most accurate localization of the pathology, for example, showing which scallop is more involved, chordae rupture, MR severity, MAD length, and 3D TEE is more accurate than 2D TEE for this purpose [31,40,41]. TEE is the gold standard for surgical planning (Figure 6).

### 4.2. Cardiac Magnetic Resonance

Cine magnetic resonance imaging is an adequate tool to study the mitral valvular apparatus as well as thickness and severity of leaflet prolapse and presence of MAD. LGE images, which are usually obtained after 10 min of gadolinium contrast injection, are excellent for the detection of areas of papillary muscle and LV fibrosis. CMR is the gold standard for LV and right ventricular volumetric assessment. CMR can also provide additional and more precise information on mitral valve characteristics, is more sensitive in MAD and curling identification [31], localized hypertrophy [14,42], and LV myocardial strain [42], although it has a poorer performance than TTE. Tissue characterization is another piece of the MVP puzzle that CMR adds. In fact, the connection between MVP with or without MAD and macro-fibrosis is well established, as described by the group of the Padua University [11,14] and then confirm by others [15,43,44,45], which involves the basal inferolateral wall (non-ischemic pattern as mid-wall striae or patchy are the most common) and papillary muscles, particularly those segments adjacent to the posteromedial papillary muscle even with only mild MR (Figure 7). Basso et al. [11] also demonstrated that VA with RBBB morphology originated from the LV inferobasal wall near the mitral annulus, where myocardial fibrosis was either detected by CMR (LGE) in patients or by histology in SCD patients.

Tissue characterization and diffuse fibrosis can be performed using T1 mapping, a technique used in various cardiomyopathies with prognostic significance. Bui at al [46] showed that patients with MVP had shorter postcontrast T1 times (higher fibrosis), particularly those with c-VA compared with patients without c-VA. In the retrospective study of Pavon et al. [43], LGE was observed in 47% of patients with MVP and MAD, but 87% of the MVP and MAD patients had an extracellular volume (ECV) above the upper limit of normal (>27%). In addition, an ECV >27% was not only found in 93% of LGE-positive patients, but also in 81% of LGE-negative patients, and ECV was increased in all myocardium adjacent to the insertion of the prolapsing valve and not only in the inferobasal and inferolateral region. The authors concluded that using LGE alone may underestimate myocardial involvement. Consistent results were reported by Guglielmo et al. [29] using native T1 mapping, showing no correlation between T1 mapping and the degree of MR. This evidence may lend support to the hypothesis that diffuse fibrosis is not a direct effect of volume overload. Interestingly, native T1 values were higher in the basal and mid-inferolateral wall than in the other segments combined with inverse strain regional distribution from the base to apex in MVP patients as compared to control subjects, suggesting a relation between mechanical deformation and tissue changes. However, no correlation between these findings and VA was reported [29]. Finally, in another study evaluating MVP patients using CMR, Daza et al. [42] found that, compared to controls, MVP patients had LV enlargement, basal inferolateral hypertrophy, increased basal longitudinal strain, particularly in the anterior, anterolateral, and inferolateral segments, and more frequently MAD, regardless of the presence of significant MR and also in the presence of borderline MVP, concluding that MVP is a complex disease that involves not only the valve, but also the LV and should be seen as a “ventriculo-mitral unit”.

CMR can help in stratifying MVP patients at risk for malignant arrhythmias as it allows the detection of either focal or diffuse fibrosis that may be responsible for reentry circuits, providing also evidence of early structural and functional remodeling that may evolve to fibrosis, although these findings need wider clinical investigation.

### 4.3. Positron Emission Tomography

It seems recognized that repetitive traction by the prolapsing leaflets and mitral annular hypermobility are responsible for local fibrosis. The hypothesis is that repetitive stretch activates fibroblasts that induce local inflammation and then fibrosis. In a pilot study using hybrid positron emission tomography (^18^F-fluorodeoxyglucose [FDG])/magnetic resonance imaging (LGE), focal ^18^F-FDG (PET positive) uptake was detected in 17 of 20 MVP patients and LGE was present in 14 of them. Patients with c-VA showed a higher LGE burden (50% with severe MR), whereas those with minor VA had higher ^18^F-FDG uptake (37.5% severe MR) [17]. The authors concluded that FDG and LGE uptake could represent different stages of the MVP disease. Further data are warranted to confirm these findings.

### 4.4. Cardiac Computed Tomography

Cardiac computed tomography (CCT) is another useful modality for the evaluation of mitral valve anatomy and MAD; however, it provides details on mitral valve anatomy, but does not allow tissue characterization or grading of MR severity and therefore it is underused. Very recently, Tho et al. [47], by evaluating the extent of disjunction using CCT, found that the anterior mitral or aortic leaflet is unaffected since it is in direct continuity with the aortic root. This study also revealed that MAD was a common finding in normal subjects. In addition, MAD had a bimodal distribution pattern in continuity with bilateral commissures of the posterior mitral leaflet, corresponding to the inferior to inferoseptal region (P3 scallop), and the anterior to anterolateral region (P1 scallop). Interestingly, the extent of MAD was greater in MVP patients than in controls (5.2–10 mm vs. 3 mm) and even greater in patients with arrhythmias compared to those without arrhythmias (8 mm vs. 6.6 mm). There are no data yet on the incremental power of CCT for risk stratification in MVP patients.

## 5. Multimodality Approach to Therapeutic Management

Multimodality imaging is useful in identifying different MVP phenotypes and can help in therapeutic decision making. TTE and TEE are the preferred imaging techniques for assessing mitral valve morphology and grade of MR severity. For FED patients with severe MR, LV, and left atrial remodeling, eventually LV dysfunction, surgery is the only option. The same approach applies to MVP patients with BD and severe MR [48,49].

The rationale for surgical intervention in MVP is to relieve papillary muscle stretching and facilitate ventricular remodeling, leading to a reduction in VA burden. There are conflicting data on the role of mitral valve repair or replacement in VA control as only case reports or small single center studies are available. Eriksson et al. [27] hypothesized that MAD could play a role in the stability and durability of annuloplasty in mitral valve repair and modified their surgical technique. Nuksuk et al. [50] revealed that, in patients with severe MR and VA, mitral valve surgery (repair in 94%) was associated with a significant reduction in VA burden in younger patients (~42 years old), but not in older ones (~62 years old), likely because older patients with long-standing MVP have more diffuse fibrosis. As a result, early surgical intervention can be more successful in terms of VA control as it is performed in less fibrotic lesions. An improvement of malignant AMVP has been reported [51,52], but opposite results have also been observed in patients operated only for arrhythmias [53].

Management of AMVP with no severe MR is more challenging and multimodality imaging could also guide AMVP intervention: hypercontractility and fibrosis could be considered as “red flags” that may provide an indication for surgery, but no data are available except for sparse case reports. Implantable cardioverter-defibrillator (ICD) therapy or electrophysiologic (EP) ablation could represent further treatment options. Risk stratification for arrhythmic events is an important component of the assessment of MVP patients and multimodality imaging plus ECG/Holter ECG and stress test are essential in this context. Patients with VA, hypercontractility on TTE (Pickelhaube sign/longitudinal strain), fibrosis on CMR should undergo additional investigation with an implantable loop recorder or electrophysiologic study. As for ICD implantation, no data exist in primary prevention. Catheter ablation can be considered when the trigger of the arrhythmia has been documented or in VT due to scar-related reentry [54]. It has been shown that ablation may not be definitive and AMVP patients may experience recurrent VT/VF, suggesting that this is a progressive disease [55]. Low-risk patients are characterized by pleomorphic PVCs, no hypercontractility on TTE, and no LGE on CMR. The management includes regular follow-up and medical therapy according to current guidelines [48,49]. Beta-blockers are the first-choice treatment for symptomatic or asymptomatic patients with non-sustained or sustained VA. Patients at high-risk for SCD and no severe MR will be considered for: (1) ICD for primary prevention (no data); (2) EP study and ablation (no data); (3) cardiac surgery (no data). In case of severe MR, cardiac surgery is the treatment of choice (Figure 8).

## 6. Conclusions

MVP is the most common mitral valve disease. Serious complications such as VA and cardiac arrest are reported. In order to understand this pathology and identify those patients at risk for serious complications, an advanced multimodality approach is required. Advanced TTE and TEE play a key role in establishing the diagnosis of MVP and assessing as well as quantifying the severity of MVP and associated regurgitation. Tissue characterization and presence of myocardial inflammation and fibrosis can help in identifying at-risk patients. CMR with and without LGE (native and post contrast T1 mapping), and maybe in the near future, ^18^F-FDG PET, are merging into the evaluation and risk stratification in this disease. MVP can present with or without MAD, and abnormal myocardial changes can be seen in both cases although MVP-MAD is more frequently associated with c-VA. Finally, multimodality imaging can guide treatment allowing the identification of different MVP phenotypes.

## Figures and Tables

**Figure 1 jcm-11-00455-f001:**
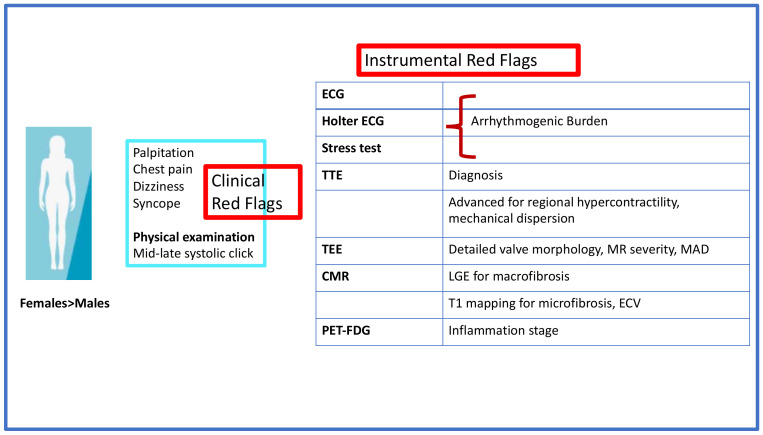
Clinical and instrumental red flags. CMR, cardiac magnetic resonance; ECG, electrocardiography; ECV, extracellular volume; FDG, fluorodeoxyglucose; LGE, late gadolinium enhancement; MAD, mitral annular disjunction; MR, mitral regurgitation; PET, positron emission tomography; TEE, transesophageal echocardiography; TTE, transthoracic echocardiography.

**Figure 2 jcm-11-00455-f002:**
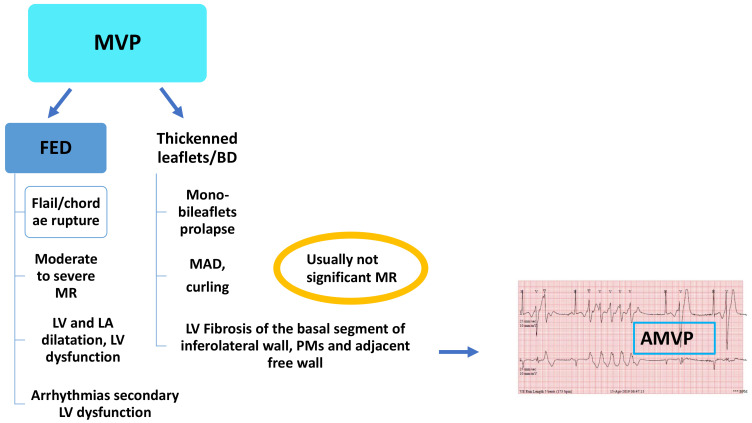
Mitral valve prolapse (MVP): fibroelastic deficiency (FED), Barlow’s disease (BD). AMVP, arrhythmic mitral valve prolapse; LA, left atrium; LV, left ventricle; MAD, mitral annulus disjunction; MR, mitral regurgitation; PM, papillary muscle.

**Figure 3 jcm-11-00455-f003:**
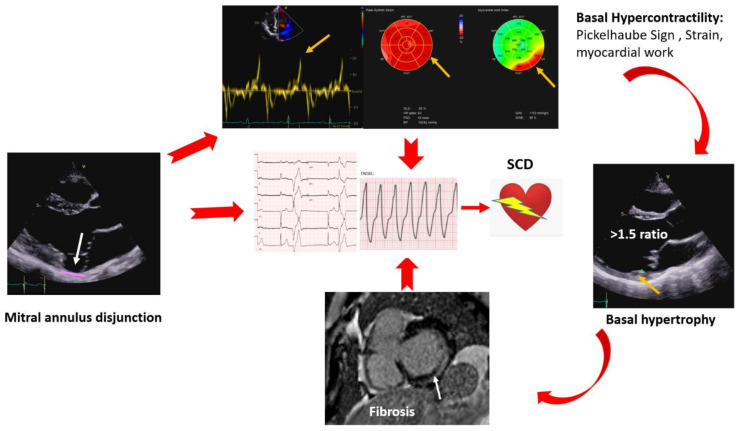
Mitral annulus disjunction, hypercontractility, increased myocardial work, basal hypertrophy, and myocardial fibrosis; all may lead to complex arrhythmias and eventually sudden cardiac death (SCD).

**Figure 4 jcm-11-00455-f004:**
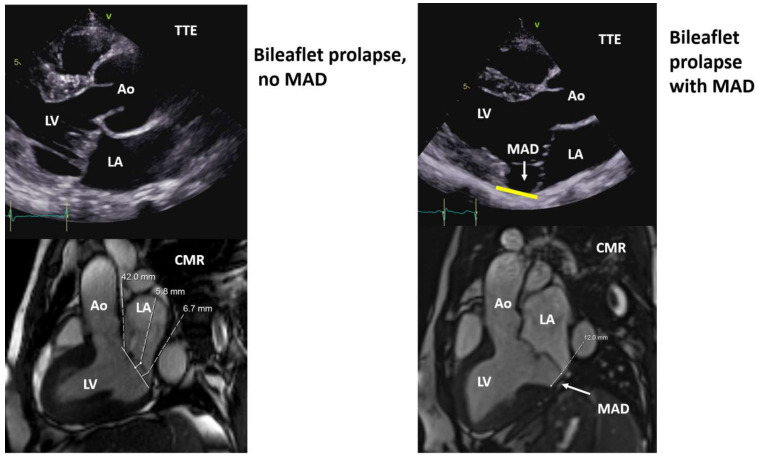
Mitral valve prolapse with no mitral annulus disjunction (MAD) and with MAD (upper panels: TTE, lower panels CMR). Ao, aorta; CMR, cardiac magnetic resonance; LA, left atrium; LV, left ventricle; TTE, transthoracic echocardiography.

**Figure 5 jcm-11-00455-f005:**
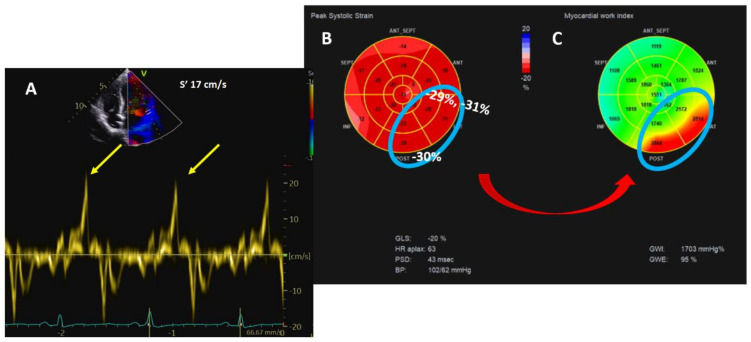
Transthoracic echocardiography parameters of hypercontractility. Panel (**A**): Pickelhaube sign. Panel (**B**): global and regional longitudinal strain. Panel (**C**): myocardial work.

**Figure 6 jcm-11-00455-f006:**
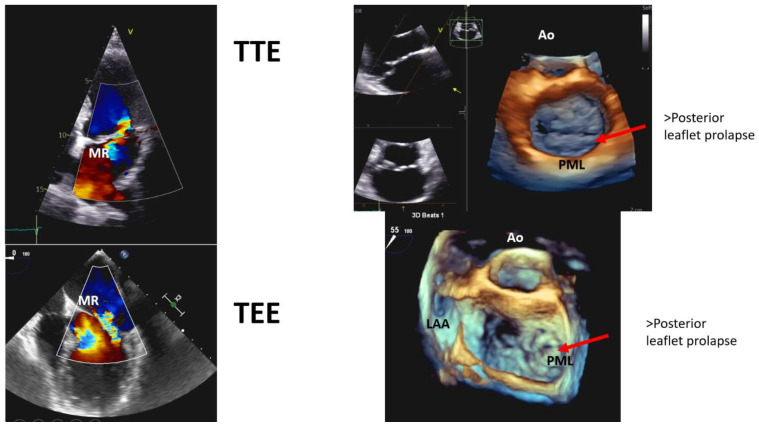
Comparison between transthoracic (TTE) and transesophageal echocardiography (TEE) in the assessment of mitral valve structure and mitral regurgitation (MR). Ao, aorta; LAA, left atrial appendage; PML, posterior mitral leaflet.

**Figure 7 jcm-11-00455-f007:**
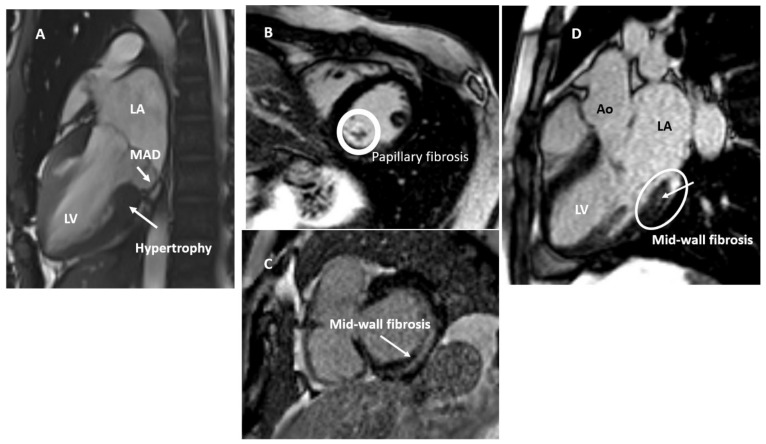
Mitral valve prolapse characteristics on cardiac magnetic resonance. Panel (**A**): basal hypertrophy in mitral valve prolapse with mitral annulus disjunction (MAD). Panel (**B**): papillary muscle fibrosis. Panel (**C**): mid-wall fibrosis (non-ischemic fibrosis). Panel (**D**): mid-wall fibrosis of the basal segment of the infero-lateral wall. Ao, aorta; LA, left atrium; LV, left ventricle.

**Figure 8 jcm-11-00455-f008:**
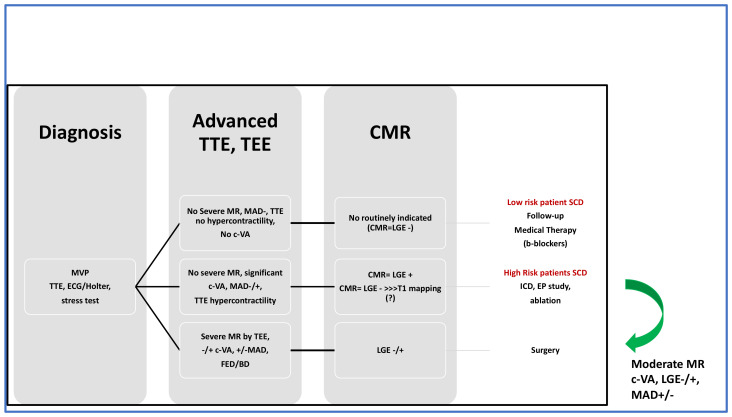
Multimodality approach for therapeutic decision making. TTE, transthoracic echocardiography; TEE, transesophageal echocardiography; CMR, cardiac magnetic resonance; ICD, implantable cardioverter-defibrillator (ICD); EP, electrophysiology study; FED, fibroelastic deficiency; BD, Barlow disease; LGE, late gadolinium enhancement; c-VA, complex ventricular arrhythmias; MAD, mitral annulus disjunction.

**Table 1 jcm-11-00455-t001:** Clinical, electrocardiographic, and multimodality imaging features of arrhythmic mitral valve prolapse.

Gender	Female
Clinical evaluation	Systolic click
ECG	T-wave inversion or biphasic wave in the inferior leads
	QT interval prolongation
	Non-specific ST changes
Holter ECG	Complex ventricular arrhythmias, bigeminy, polymorphic, RBBB (most of the cases)
TTE findings	Bileaflet prolapse
	Myxomatous degeneration
	Moderate to severe MR (only in 36% of cases)
	Mitral annulus disjunction
	Curling of the mitral annulus
	High lateral S’ on TDI (>16 cm/s) (Pickelhaube sign)
	Basal/mid-segment inferolateral wall thickness ratio >1.5
	Paradoxical movement of the mitral annulus
Advanced TTE	Prolonged mechanical dispersion
	Impaired global longitudinal strain
	Increased myocardial work
TEE	Additional information on MR severity
	Detailed mitral valve anatomy
CMR	LGE (macrofibrosis)
	Fibrosis replacement in the basal and mid segments of the inferolateral and inferobasal wall
	T1 mapping (microfibrosis)
	Diffuse fibrosis: T1 relaxation higher than controls; T1 relaxation similar to MVP–MAD patients with positive LGE
	ECV increased in LGE-negative patients and can be similar to LGE-positive MVP patients
Positron emission tomography	Surrogate for myocardial inflammation Less FDG uptake in c-VAs (less inflammation and more fibrosis) compared with moderate VAs

CMR, cardiac magnetic resonance; ECV, extracellular volume; LGE, late gadolinium enhancement; MAD, mitral annulus disjunction; MR, mitral regurgitation; MVP, mitral valve prolapse; PET, positron emission tomography; RBBB, right bundle branch block; TDI, tissue Doppler imaging; TEE, transesophageal echocardiography; TTE, transthoracic echocardiography; c-VA, complex ventricular arrhythmia.

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
