# Peer review of "Diagnosis of Mitral Valve Prolapse: Much More than Simple Prolapse. Multimodality Approach to Risk Stratification and Therapeutic Management"

_jcm, 2022, doi:10.3390/jcm11020455_

Round 1
Reviewer 1 Report
This article discusses an interesting subject. I have some suggestions for the authors.
- The authors state that “Mitral valve prolapse (MVP) is the displacement of one or both mitral leaflets into 35 the left atrium in systole,” which is not entirely correct, since in most cases only 1 scallop of the leaflet can be involved, not the entire leaflet.
- In Line 39- the authors say that MVP is rare in young adults. The term “young adults” is to general, especially when referring to MVP which occurs in younger people than other cardiovascular diseases. The authors should specify this age group?
- Line 46 - The authors mention MVP as a cause of endocarditis. The probability is very small.
- Line 334- This is this the Legend of figure 8? In this case, it should be under the figure, not after another paragraph.
Author Response
- The authors state that “Mitral valve prolapse (MVP) is the displacement of one or both mitral leaflets into the left atrium in systole,” which is not entirely correct, since in most cases only 1 scallop of the leaflet can be involved, not the entire leaflet.
We thank the reviewer for the puntualization. We agree that most of the time the prolapse involves part of the leaflet (scallop) and, as suggested, the definition has been corrected.
- In Line 39- the authors say that MVP is rare in young adults. The term “young adults” is to general, especially when referring to MVP which occurs in younger people than other cardiovascular diseases. The authors should specify this age group?
As suggested we changed “young adults“ to adolescent that was what we meant and what it was reported in literature.
- Line 46 - The authors mention MVP as a cause of endocarditis. The probability is very small.
We agree that endocarditis is quite rare. It was mentioned as completeness. We modify the sentence accordingly.
- Line 334- This is this the Legend of figure 8? In this case, it should be under the figure, not after another paragraph.
The legend was place under Fig 8 as suggested and paragraph corresponding to line 342-344 was re-adjusted accordingly.

Reviewer 2 Report
Alenazy et al. present a comprehensive review on mitral valve prolaps summing up the recent knowledge about disease definition, diagnostics, imaging and its implications on the developement of ventricular arrhythmias.
The review is well written and structured.
I have no major issues with the manuscript and want to congratulate the authors for their work.
Author Response
The review is well written and structured.
I have no major issues with the manuscript and want to congratulate the authors for their work.
We thank the reviewer for the comment
